# OFFLINE REINFORCEMENT LEARNING WITH DIFFERENTIAL PRIVACY

## ABSTRACT

The offline reinforcement learning (RL) problem is often motivated by the need to learn data-driven decision policies in financial, legal and healthcare applications. However, the learned policy could retain sensitive information of individuals in the training data (e.g., treatment and outcome of patients), thus susceptible to various privacy risks. We design offline RL algorithms with differential privacy guarantees which provably prevent such risks. These algorithms also enjoy strong instance-dependent learning bounds under both tabular and linear Markov Decision Process (MDP) settings. Our theory and simulation suggest that the privacy guarantee comes at (almost) no drop in utility comparing to the non-private counterpart for a medium-size dataset.

## 1 INTRODUCTION

Offline Reinforcement Learning (or batch RL) aims to learn a near-optimal policy in an unknown environment[1] through a static dataset gathered from some behavior policy $\mu$. Since offline RL does not require access to the environment, it can be applied to problems where interaction with environment is infeasible, *e.g.*, when collecting new data is costly (trade or finance (Zhang et al., 2020)), risky (autonomous driving (Sallab et al., 2017)) or illegal / unethical (healthcare (Raghu et al., 2017)). In such practical applications, the data used by an RL agent usually contains sensitive information. Take medical history for instance, for each patient, at each time step, the patient reports her health condition (age, disease, etc.), then the doctor decides the treatment (which medicine to use, the dosage of medicine, etc.), finally there is treatment outcome (whether the patient feels good, etc.) and the patient transitions to another health condition. Here, (health condition, treatment, treatment outcome) corresponds to (state, action, reward) and the dataset can be considered as $n$ (number of patients) trajectories sampled from a MDP with horizon $H$ (number of treatment steps). However, learning agents are known to implicitly memorize details of individual training data points verbatim (Carlini et al., 2019), even if they are irrelevant for learning (Brown et al., 2021), which makes offline RL models vulnerable to various privacy attacks.

Differential privacy (DP) (Dwork et al., 2006) is a well-established definition of privacy with many desirable properties. A differentially private offline RL algorithm will return a decision policy that is indistinguishable from a policy trained in an alternative universe any individual user is replaced, thereby preventing the aforementioned privacy risks. There is a surge of recent interest in developing RL algorithms with DP guarantees, but they focus mostly on the online setting (Vietri et al., 2020; Garcelon et al., 2021; Liao et al., 2021; Chowdhury & Zhou, 2021; Luyo et al., 2021).

Offline RL is arguably more practically relevant than online RL in the applications with sensitive data. For example, in the healthcare domain, online RL requires actively running new exploratory policies (clinical trials) with every new patient, which often involves complex ethical / legal clearances, whereas offline RL uses only historical patient records that are often accessible for research purposes. Clear communication of the adopted privacy enhancing techniques (e.g., DP) to patients was reported to further improve data access (Kim et al., 2017).

**Our contributions.** In this paper, we present the first provably efficient algorithms for offline RL with differential privacy. Our contributions are twofold.

---

[1]The environment is usually characterized by a Markov Decision Process (MDP) in this paper.

- We design two new pessimism-based algorithms DP-APVI (Algorithm 1) and DP-VAPVI (Algorithm 2), one for the tabular setting (finite states and actions), the other for the case with linear function approximation (under linear MDP assumption). Both algorithms enjoy DP guarantees (pure DP or zCDP) and instance-dependent learning bounds where the cost of privacy appears as lower order terms.
- We perform numerical simulations to evaluate and compare the performance of our algorithm DP-VAPVI (Algorithm 2) with its non-private counterpart VAPVI (Yin et al., 2022) as well as a popular baseline PEVI (Jin et al., 2021). The results complement the theoretical findings by demonstrating the practicality of DP-VAPVI under strong privacy parameters.

**Related work.** To our knowledge, differential privacy in offline RL tasks has not been studied before, except for much simpler cases where the agent only evaluates a single policy (Balle et al., 2016; Xie et al., 2019). Balle et al. (2016) privatized first-visit Monte Carlo-Ridge Regression estimator by an output perturbation mechanism and Xie et al. (2019) used DP-SGD. Neither paper considered offline learning (or policy optimization), which is our focus.

There is a larger body of work on private RL in the online setting, where the goal is to minimize regret while satisfying either joint differential privacy (Vietri et al., 2020; Chowdhury & Zhou, 2021; Ngo et al., 2022; Luyo et al., 2021) or local differential privacy (Garcelon et al., 2021; Liao et al., 2021; Luyo et al., 2021; Chowdhury & Zhou, 2021). The offline setting introduces new challenges in DP as we cannot *algorithmically enforce* good "exploration", but have to work with a static dataset and privately estimate the uncertainty in addition to the value functions. A private online RL algorithm can sometimes be adapted for private offline RL too, but those from existing work yield suboptimal and non-adaptive bounds. We give a more detailed technical comparison in Appendix B.

Among non-private offline RL works, we build directly upon non-private offline RL methods known as Adaptive Pessimistic Value Iteration (APVI, for tabular MDPs) (Yin & Wang, 2021b) and Variance-Aware Pessimistic Value Iteration (VAPVI, for linear MDPs) (Yin et al., 2022), as they give the strongest theoretical guarantees to date. We refer readers to Appendix B for a more extensive review of the offline RL literature. Introducing DP to APVI and VAPVI while retaining the same sample complexity (modulo lower order terms) require nontrivial modifications to the algorithms.

**A remark on technical novelty.** Our algorithms involve substantial technical innovation over previous works on online DP-RL with joint DP guarantee[2]. Different from previous works, our DP-APVI (Algorithm 1) operates on Bernstein type pessimism, which requires our algorithm to deal with conditional variance using private statistics. Besides, our DP-VAPVI (Algorithm 2) replaces the LSVI technique with variance-aware LSVI (also known as weighted ridge regression, first appears in (Zhou et al., 2021)). Our DP-VAPVI releases conditional variance privately, and further applies weighted ridge regression privately. Both approaches ensure tighter instance-dependent bounds on the suboptimality of the learned policy.

## 2 PROBLEM SETUP

**Markov Decision Process.** A finite-horizon *Markov Decision Process* (MDP) is denoted by a tuple $M = (\mathcal{S}, \mathcal{A}, P, r, H, d_1)$ (Sutton & Barto, 2018), where $\mathcal{S}$ is state space and $\mathcal{A}$ is action space. A non-stationary transition kernel $P_h : \mathcal{S} \times \mathcal{A} \times \mathcal{S} \mapsto [0, 1]$ maps each state action $(s_h, a_h)$ to a probability distribution $P_h(\cdot|s_h, a_h)$ and $P_h$ can be different across time. Besides, $r_h : \mathcal{S} \times \mathcal{A} \mapsto \mathbb{R}$ is the expected immediate reward satisfying $0 \le r_h \le 1$, $d_1$ is the initial state distribution and $H$ is the horizon. A policy $\pi = (\pi_1, \cdots, \pi_H)$ assigns each state $s_h \in \mathcal{S}$ a probability distribution over actions according to the map $s_h \mapsto \pi_h(\cdot|s_h), \forall h \in [H]$. A random trajectory $s_1, a_1, r_1, \cdots, s_H, a_H, r_H, s_{H+1}$ is generated according to $s_1 \sim d_1, a_h \sim \pi_h(\cdot|s_h), r_h \sim r_h(s_h, a_h), s_{h+1} \sim P_h(\cdot|s_h, a_h), \forall h \in [H]$.

For tabular MDP, we have $\mathcal{S} \times \mathcal{A}$ is the discrete state-action space and $S := |\mathcal{S}|, A := |\mathcal{A}|$ are finite. In this work, we assume that $r$ is known[3]. In addition, we denote the per-step marginal state-action occupancy $d_h^\pi(s, a)$ as: $d_h^\pi(s, a) := \mathbb{P}[s_h = s|s_1 \sim d_1, \pi] \cdot \pi_h(a|s)$, which is the marginal state-action probability at time $h$.

---

[2]Here we only compare our techniques (for offline RL) with the works for online RL under joint DP guarantee, as both settings allow access to the raw data.

[3]This is due to the fact that the uncertainty of reward function is dominated by that of transition kernel in RL.

**Value function, Bellman (optimality) equations.** The value function $V_h^\pi(\cdot)$ and Q-value function $Q_h^\pi(\cdot, \cdot)$ for any policy $\pi$ is defined as: $V_h^\pi(s) = \mathbb{E}_\pi[\sum_{t=h}^H r_t | s_h = s]$, $Q_h^\pi(s, a) = \mathbb{E}_\pi[\sum_{t=h}^H r_t | s_h = s, a_h = s, a]$, $\forall h, s, a \in [H] \times \mathcal{S} \times \mathcal{A}$. The performance is defined as $v^\pi := \mathbb{E}_{d_1}[V_1^\pi] = \mathbb{E}_{\pi, d_1}\left[\sum_{t=1}^H r_t\right]$. The Bellman (optimality) equations follow $\forall h \in [H]$: $Q_h^\pi = r_h + P_h V_{h+1}^\pi$, $V_h^\pi = \mathbb{E}_{a \sim \pi_h}[Q_h^\pi]$, $Q_h^\star = r_h + P_h V_{h+1}^\star$, $V_h^\star = \max_a Q_h^\star(\cdot, a)$.

**Linear MDP (Jin et al., 2020b).** An episodic MDP $(\mathcal{S}, \mathcal{A}, H, P, r)$ is called a linear MDP with known feature map $\phi : \mathcal{S} \times \mathcal{A} \to \mathbb{R}^d$ if there exist $H$ unknown signed measures $\nu_h \in \mathbb{R}^d$ over $\mathcal{S}$ and $H$ unknown reward vectors $\theta_h \in \mathbb{R}^d$ such that

$$P_h(s' \mid s, a) = \langle \phi(s, a), \nu_h(s') \rangle, \quad r_h(s, a) = \langle \phi(s, a), \theta_h \rangle, \quad \forall (h, s, a, s') \in [H] \times \mathcal{S} \times \mathcal{A} \times \mathcal{S}.$$

Without loss of generality, we assume $\|\phi(s, a)\|_2 \le 1$ and $\max(\|\nu_h(\mathcal{S})\|_2, \|\theta_h\|_2) \le \sqrt{d}$ for all $h, s, a \in [H] \times \mathcal{S} \times \mathcal{A}$. An important property of linear MDP is that the value functions are linear in the feature map, which is summarized in Lemma E.14.

**Offline setting and the goal.** The offline RL requires the agent to find a policy $\pi$ in order to maximize the performance $v^\pi$, given only the episodic data $\mathcal{D} = \{(s_h^\tau, a_h^\tau, r_h^\tau, s_{h+1}^\tau)\}_{\tau \in [n]}^{h \in [H]}$[4] rolled out from some fixed and possibly unknown behavior policy $\mu$, which means we cannot change $\mu$ and in particular we do not assume the functional knowledge of $\mu$. In conclusion, based on the batch data $\mathcal{D}$ and a targeted accuracy $\epsilon > 0$, the agent seeks to find a policy $\pi_{\text{alg}}$ such that $v^\star - v^{\pi_{\text{alg}}} \le \epsilon$.

## 2.1 ASSUMPTIONS IN OFFLINE RL

In order to show that our privacy-preserving algorithms can generate near optimal policy, certain coverage assumptions are needed. In this section, we will list the assumptions we use in this paper.

**Assumptions for tabular setting.**

**Assumption 2.1** ((Liu et al., 2019)). *There exists one optimal policy $\pi^\star$, such that $\pi^\star$ is fully covered by $\mu$, i.e. $\forall s_h, a_h \in \mathcal{S} \times \mathcal{A}$, $d_h^{\pi^\star}(s_h, a_h) > 0$ only if $d_h^\mu(s_h, a_h) > 0$. Furthermore, we denote the trackable set as $\mathcal{C}_h := \{(s_h, a_h) : d_h^\mu(s_h, a_h) > 0\}$.*

Assumption 2.1 is the weakest assumption needed for accurately learning the optimal value $v^\star$ by requiring $\mu$ to trace the state-action space of one optimal policy ($\mu$ can be agnostic at other locations). Similar to (Yin & Wang, 2021b), we will use Assumption 2.1 for the tabular part of this paper, which enables comparison between our sample complexity to the conclusion in (Yin & Wang, 2021b), whose algorithm serves as a non-private baseline.

**Assumptions for linear setting.** First, we define the expectation of covariance matrix under the behavior policy $\mu$ for all time step $h \in [H]$ as below:

$$\Sigma_h^p := \mathbb{E}_\mu \left[ \phi(s_h, a_h) \phi(s_h, a_h)^\top \right]. \tag{1}$$

As have been shown in (Wang et al., 2021; Yin et al., 2022), learning a near-optimal policy from offline data requires coverage assumptions. Here in linear setting, such coverage is characterized by the minimum eigenvalue of $\Sigma_h^p$. Similar to (Yin et al., 2022), we apply the following assumption for the sake of comparison.

**Assumption 2.2** (Feature Coverage, Assumption 2 in (Wang et al., 2021)). *The data distributions $\mu$ satisfy the minimum eigenvalue condition: $\forall h \in [H]$, $\kappa_h := \lambda_{\min}(\Sigma_h^p) > 0$. Furthermore, we denote $\kappa = \min_h \kappa_h$.*

## 2.2 DIFFERENTIAL PRIVACY IN OFFLINE RL

In this work, we aim to design privacy-preserving algorithms for offline RL. We apply differential privacy as the formal notion of privacy. Below we revisit the definition of differential privacy.

**Definition 2.3** (Differential Privacy (Dwork et al., 2006)). *A randomized mechanism $M$ satisfies $(\epsilon, \delta)$-differential privacy $((\epsilon, \delta)$-DP) if for all neighboring datasets $U, U'$ that differ by one data point and for all possible event $E$ in the output range, it holds that*

$$\mathbb{P}[M(U) \in E] \le e^\epsilon \cdot \mathbb{P}[M(U') \in E] + \delta.$$

---

[4]For clarity we use $n$ for tabular MDP and $K$ for linear MDP when referring to the sample complexity.

*When $\delta = 0$, we say pure DP, while for $\delta > 0$, we say approximate DP.*

In the problem of offline RL, the dataset consists of several trajectories, therefore one data point in Definition 2.3 refers to one single trajectory. Hence the definition of Differential Privacy means that the difference in the distribution of the output policy resulting from replacing one trajectory in the dataset will be small. In other words, an adversary can not infer much information about any single trajectory in the dataset from the output policy of the algorithm.

During the whole paper, we will use zCDP (defined below) as a surrogate for DP, since it enables cleaner analysis for privacy composition and Gaussian mechanism. The properties of zCDP (e.g., composition, conversion formula to DP) are deferred to Appendix E.3.

**Definition 2.4** (zCDP (Dwork & Rothblum, 2016; Bun & Steinke, 2016)). *A randomized mechanism $M$ satisfies $\rho$-Zero-Concentrated Differential Privacy ($\rho$-zCDP), if for all neighboring datasets $U, U'$ and all $\alpha \in (1, \infty)$,*

$$D_\alpha(M(U)\|M(U')) \leq \rho\alpha,$$

*where $D_\alpha$ is the Renyi-divergence (Van Erven & Harremos, 2014).*

Finally, we go over the definition and privacy guarantee of Gaussian mechanism.

**Definition 2.5** (Gaussian Mechanism (Dwork et al., 2014)). *Define the $\ell_2$ sensitivity of a function $f : \mathbb{N}^{\mathcal{X}} \mapsto \mathbb{R}^d$ as*

$$\Delta_2(f) = \sup_{neighboring\ U, U'} \|f(U) - f(U')\|_2.$$

*The Gaussian mechanism $\mathcal{M}$ with noise level $\sigma$ is then given by*

$$\mathcal{M}(U) = f(U) + \mathcal{N}(0, \sigma^2 I_d).$$

**Lemma 2.6** (Privacy guarantee of Gaussian mechanism (Dwork et al., 2014; Bun & Steinke, 2016)). *Let $f : \mathbb{N}^{\mathcal{X}} \mapsto \mathbb{R}^d$ be an arbitrary d-dimensional function with $\ell_2$ sensitivity $\Delta_2$. Then for any $\rho > 0$, Gaussian Mechanism with parameter $\sigma^2 = \frac{\Delta_2^2}{2\rho}$ satisfies $\rho$-zCDP. In addition, for all $0 < \delta, \epsilon < 1$, Gaussian Mechanism with parameter $\sigma = \frac{\Delta_2}{\epsilon}\sqrt{2\log\frac{1.25}{\delta}}$ satisfies $(\epsilon, \delta)$-DP.*

We emphasize that the privacy guarantee covers any input data. It does *not* require any distributional assumptions on the data. The RL-specific assumptions (e.g., linear MDP and coverage assumptions) are only used for establishing provable utility guarantees.

## 3 RESULTS UNDER TABULAR MDP: DP-APVI (ALGORITHM 1)

For reinforcement learning, the tabular MDP setting is the most well-studied setting and our first result applies to this regime. We begin with the construction of private counts.

**Private Model-based Components.** Given data $\mathcal{D} = \{(s_h^\tau, a_h^\tau, r_h^\tau, s_{h+1}^\tau)\}_{\tau \in [n]}^{h \in [H]}$, we denote $n_{s_h, a_h} := \sum_{\tau=1}^n \mathbb{1}[s_h^\tau, a_h^\tau = s_h, a_h]$ be the total counts that visit $(s_h, a_h)$ pair at time $h$ and $n_{s_h, a_h, s_{h+1}} := \sum_{\tau=1}^n \mathbb{1}[s_h^\tau, a_h^\tau, s_{h+1}^\tau = s_h, a_h, s_{h+1}]$ be the total counts that visit $(s_h, a_h, s_{h+1})$ pair at time $h$, then given the budget $\rho$ for zCDP, we add *independent* Gaussian noises to all the counts:

$$n'_{s_h, a_h} = \{n_{s_h, a_h} + \mathcal{N}(0, \sigma^2)\}^+, \ n'_{s_h, a_h, s_{h+1}} = \{n_{s_h, a_h, s_{h+1}} + \mathcal{N}(0, \sigma^2)\}^+, \ \sigma^2 = \frac{2H}{\rho}. \quad (2)$$

However, after adding noise, the noisy counts $n'$ may not satisfy $n'_{s_h, a_h} = \sum_{s_{h+1} \in \mathcal{S}} n'_{s_h, a_h, s_{h+1}}$. To address this problem, we choose the private counts of visiting numbers as the solution to the following optimization problem (here $E_\rho = 4\sqrt{\frac{H\log\frac{4HS^2A}{\delta}}{\rho}}$):

$$\{\widetilde{n}_{s_h, a_h, s'}\}_{s' \in \mathcal{S}} = \operatorname{argmin}_{\{x_{s'}\}_{s' \in \mathcal{S}}} \max_{s' \in \mathcal{S}} |x_{s'} - n'_{s_h, a_h, s'}|$$

$$\text{such that } \left|\sum_{s' \in \mathcal{S}} x_{s'} - n'_{s_h, a_h}\right| \leq \frac{E_\rho}{2} \text{ and } x_{s'} \geq 0, \forall s' \in \mathcal{S}. \quad (3)$$

$$\widetilde{n}_{s_h, a_h} = \sum_{s' \in \mathcal{S}} \widetilde{n}_{s_h, a_h, s'}.$$

**Remark 3.1.** *The optimization problem* (3) *can be reformulated as:*

$$\min \ t, \ s.t. \ |x_{s'} - n'_{s_h,a_h,s'}| \le t \ and \ x_{s'} \ge 0 \ \forall s' \in \mathcal{S}, \ \left| \sum_{s' \in \mathcal{S}} x_{s'} - n'_{s_h,a_h} \right| \le \frac{E_\rho}{2}. \quad (4)$$

*Note that* (4) *is a* Linear Programming *problem with $S+1$ variables and $2S+2$ (one constraint on absolute value is equivalent to two linear constraints) linear constraints, which can be solved efficiently by the simplex method (Ficken, 2015) or other provably efficient algorithms (Nemhauser & Wolsey, 1988). In addition, if we do not solve this optimization problem and directly take $\widetilde{n}_{s_h,a_h,s_{h+1}} = n'_{s_h,a_h,s_{h+1}}$ and $\widetilde{n}_{s_h,a_h} = \sum_{s_{h+1} \in \mathcal{S}} \widetilde{n}_{s_h,a_h,s_{h+1}}$, we can only derive $|\widetilde{n}_{s_h,a_h} - n_{s_h,a_h}| \le \widetilde{O}(\sqrt{S}E_\rho)$ through concentration on summation of S i.i.d. Gaussian noises. In contrast, solving* (3) *ensures that $|\widetilde{n}_{s_h,a_h} - n_{s_h,a_h}| \le E_\rho$ with high probability* [5].

The private estimation of the transition kernel is defined as:

$$\widetilde{P}_h(s'|s_h,a_h) = \frac{\widetilde{n}_{s_h,a_h,s'}}{\widetilde{n}_{s_h,a_h}}, \quad (5)$$

if $\widetilde{n}_{s_h,a_h} > E_\rho$ and $\widetilde{P}_h(s'|s_h,a_h) = \frac{1}{S}$ otherwise.

**Remark 3.2.** *Different from the transition kernel estimate in previous works (Vietri et al., 2020; Chowdhury & Zhou, 2021) that may not be a distribution, we have to ensure that ours is a probability distribution, because our Bernstein type pessimism (line 5 in Algorithm 1) needs to take variance over this transition kernel estimate. The intuition behind the construction of our private transition kernel is that, for those state-action pairs with $\widetilde{n}_{s_h,a_h} \le E_\rho$, we can not distinguish whether the non-zero private count comes from noise or actual visitation. Therefore we only take the empirical estimate of the state-action pairs with sufficiently large $\widetilde{n}_{s_h,a_h}$.*

---

**Algorithm 1** Differentially Private Adaptive Pessimistic Value Iteration (DP-APVI)

---

1: **Input:** Offline dataset $\mathcal{D} = \{(s_h^\tau, a_h^\tau, r_h^\tau, s_{h+1}^\tau)\}_{\tau,h=1}^{n,H}$. Reward function $r$. Constants $C_1 = \sqrt{2}, C_2 = 16, C > 1$, failure probability $\delta$, budget for zCDP $\rho$.
2: **Initialization:** Calculate $\widetilde{n}_{s_h,a_h}, \widetilde{n}_{s_h,a_h,s_{h+1}}$ as (3), $\widetilde{P}_h(s_{h+1}|s_h,a_h)$ as (5). $\widetilde{V}_{H+1}(\cdot) \leftarrow 0$. $E_\rho \leftarrow 4\sqrt{\frac{H \log \frac{4HS^2A}{\delta}}{\rho}}$. $\iota \leftarrow \log(HSA/\delta)$.
3: **for** $h = H, H-1, \ldots, 1$ **do**
4:     $\widetilde{Q}_h(\cdot, \cdot) \leftarrow r_h(\cdot, \cdot) + (\widetilde{P}_h \cdot \widetilde{V}_{h+1})(\cdot, \cdot)$
5:     $\forall s_h, a_h$, let $\Gamma_h(s_h, a_h) \leftarrow C_1 \sqrt{\frac{\text{Var}_{\widetilde{P}_{s_h,a_h}}(\widetilde{V}_{h+1}) \cdot \iota}{\widetilde{n}_{s_h,a_h} - E_\rho}} + \frac{C_2 SHE_\rho \cdot \iota}{\widetilde{n}_{s_h,a_h}}$ if $\widetilde{n}_{s_h,a_h} > E_\rho$, otherwise $CH$.
6:     $\widehat{Q}_h^p(\cdot, \cdot) \leftarrow \widetilde{Q}_h(\cdot, \cdot) - \Gamma_h(\cdot, \cdot)$.
7:     $\overline{Q}_h(\cdot, \cdot) \leftarrow \min\{\widehat{Q}_h^p(\cdot, \cdot), H-h+1\}^+$.
8:     $\forall s_h$, let $\widehat{\pi}_h(\cdot|s_h) \leftarrow \text{argmax}_{\pi_h} \langle \overline{Q}_h(s_h, \cdot), \pi_h(\cdot|s_h) \rangle$ and $\widetilde{V}_h(s_h) \leftarrow \langle \overline{Q}_h(s_h, \cdot), \widehat{\pi}_h(\cdot|s_h) \rangle$.
9: **end for**
10: **Output:** $\{\widehat{\pi}_h\}$.

---

**Algorithmic design.** Our algorithmic design originates from the idea of pessimism, which holds conservative view towards the locations with high uncertainty and prefers the locations we have more confidence about. Based on the Bernstein type pessimism in APVI (Yin & Wang, 2021b), we design a similar pessimistic algorithm with private counts to ensure differential privacy. If we replace $\widetilde{n}$ and $\widetilde{P}$ with $n$ and $\widehat{P}$[6], then our DP-APVI (Algorithm 1) will degenerate to APVI. Compared to the pessimism defined in APVI, our pessimistic penalty has an additional term $\widetilde{O}\left(\frac{SHE_\rho}{\widetilde{n}_{s_h,a_h}}\right)$, which accounts for the additional pessimism due to our application of private statistics.

We state our main theorem about DP-APVI below, the proof sketch is deferred to Appendix C.1 and detailed proof is deferred to Appendix C due to space limit.

---

[5]This conclusion is summarized in Lemma C.3.
[6]The non-private empirical estimate, defined as (15) in Appendix C.

**Theorem 3.3.** *DP-APVI (Algorithm 1) satisfies $\rho$-zCDP. Furthermore, under Assumption 2.1, denote $\bar{d}_m := \min_{h \in [H]}\{d_h^\mu(s_h, a_h) : d_h^\mu(s_h, a_h) > 0\}$. For any $0 < \delta < 1$, there exists constant $c_1 > 0$, such that when $n > c_1 \cdot \max\{H^2, E_\rho\}/\bar{d}_m \cdot \iota$ ($\iota = \log(HSA/\delta)$), with probability $1 - \delta$, the output policy $\widehat{\pi}$ of DP-APVI satisfies ($\widetilde{O}$ hides constants and Polylog terms, $E_\rho = 4\sqrt{\frac{H\log\frac{4HS^2A}{\delta}}{\rho}}$)*

$$0 \leq v^\star - v^{\widehat{\pi}} \leq 4\sqrt{2}\sum_{h=1}^{H}\sum_{(s_h,a_h)\in\mathcal{C}_h}d_h^{\pi^\star}(s_h, a_h)\sqrt{\frac{\text{Var}_{P_h(\cdot|s_h,a_h)}(V_{h+1}^\star(\cdot))\cdot\iota}{nd_h^\mu(s_h, a_h)}} + \widetilde{O}\left(\frac{H^3 + SH^2E_\rho}{n\cdot\bar{d}_m}\right). \tag{6}$$

**Comparison to non-private counterpart APVI (Yin & Wang, 2021b).** According to Theorem 4.1 in (Yin & Wang, 2021b), the sub-optimality bound of APVI is for large enough $n$, with high probability, the output $\widehat{\pi}$ satisfies:

$$0 \leq v^\star - v^{\widehat{\pi}} \leq \widetilde{O}\left(\sum_{h=1}^{H}\sum_{(s_h,a_h)\in\mathcal{C}_h}d_h^{\pi^\star}(s_h, a_h)\sqrt{\frac{\text{Var}_{P_h(\cdot|s_h,a_h)}(V_{h+1}^\star(\cdot))}{nd_h^\mu(s_h, a_h)}}\right) + \widetilde{O}\left(\frac{H^3}{n\cdot\bar{d}_m}\right). \tag{7}$$

Compared to our Theorem 3.3, the additional sub-optimality bound due to differential privacy is $\widetilde{O}\left(\frac{SH^2E_\rho}{n\cdot\bar{d}_m}\right) = \widetilde{O}\left(\frac{SH^{\frac{5}{2}}}{n\cdot\bar{d}_m\sqrt{\rho}}\right) = \widetilde{O}\left(\frac{SH^{\frac{5}{2}}}{n\cdot\bar{d}_m\epsilon}\right)$[7] In the most popular regime where the privacy budget $\rho$ or $\epsilon$ is a constant, the additional term due to differential privacy appears as a lower order term, hence becomes negligible as the sample complexity $n$ becomes large.

**Comparison to Hoeffding type pessimism.** We can simply revise our algorithm by using Hoeffding type pessimism, which replaces the pessimism in line 5 with $C_1 H \cdot \sqrt{\frac{\iota}{\widetilde{n}_{s_h,a_h}-E_\rho}} + \frac{C_2SHE_\rho\cdot\iota}{\widetilde{n}_{s_h,a_h}}$. Then with a similar proof schedule, we can arrive at a sub-optimality bound that with high probability,

$$0 \leq v^\star - v^{\widehat{\pi}} \leq \widetilde{O}\left(H \cdot \sum_{h=1}^{H}\sum_{(s_h,a_h)\in\mathcal{C}_h}d_h^{\pi^\star}(s_h, a_h)\sqrt{\frac{1}{nd_h^\mu(s_h, a_h)}}\right) + \widetilde{O}\left(\frac{SH^2E_\rho}{n\cdot\bar{d}_m}\right). \tag{8}$$

Compared to our Theorem 3.3, our bound is tighter because we express the dominate term by the system quantities instead of explicit dependence on $H$ (and $\text{Var}_{P_h(\cdot|s_h,a_h)}(V_{h+1}^\star(\cdot)) \leq H^2$). In addition, we highlight that according to Theorem G.1 in (Yin & Wang, 2021b), our main term nearly matches the non-private minimax lower bound. For more detailed discussions about our main term and how it subsumes other optimal learning bounds, we refer readers to (Yin & Wang, 2021b).

**Apply Laplace Mechanism to achieve pure DP.** To achieve Pure DP instead of $\rho$-zCDP, we can simply replace Gaussian Mechanism with Laplace Mechanism (defined as Definition E.19). Given privacy budget for Pure DP $\epsilon$, since the $\ell_1$ sensitivity of $\{n_{s_h,a_h}\} \cup \{n_{s_h,a_h,s_{h+1}}\}$ is $\Delta_1 = 4H$, we can add *independent* Laplace noises $\text{Lap}(\frac{4H}{\epsilon})$ to each count to achieve $\epsilon$-DP due to Lemma E.20. Then by using $E_\epsilon = \widetilde{O}\left(\frac{H}{\epsilon}\right)$ instead of $E_\rho$ and keeping everything else ((3), (5) and Algorithm 1) the same, we can reach a similar result to Theorem 3.3 with the same proof schedule. The only difference is that here the additional learning bound is $\widetilde{O}\left(\frac{SH^3}{n\cdot\bar{d}_m\epsilon}\right)$, which still appears as a lower order term.

## 4 RESULTS UNDER LINEAR MDP: DP-VAPVI(ALGORITHM 2)

In large MDPs, to address the computational issues, the technique of function approximation is widely applied, and linear MDP is a concrete model to study linear function approximations. Our second result applies to the linear MDP setting. Generally speaking, function approximation reduces the dimensionality of private releases comparing to the tabular MDPs. We begin with private counts.

**Private Model-based Components.** Given the two datasets $\mathcal{D}$ and $\mathcal{D}'$ (both from $\mu$) as in Algorithm 2, we can apply variance-aware pessimistic value iteration to learn a near optimal policy as in

---

[7]Here we apply the second part of Lemma 2.6 to achieve $(\epsilon, \delta)$-DP, the notation $\widetilde{O}$ also absorbs $\log\frac{1}{\delta}$ (only here $\delta$ denotes the privacy budget instead of failure probability).

VAPVI (Yin et al., 2022). To ensure differential privacy, we add *independent* Gaussian noises to the $5H$ statistics as in DP-VAPVI (Algorithm 2) below. Since there are $5H$ statistics, by the adaptive composition of zCDP (Lemma E.17), it suffices to keep each count $\rho_0$-zCDP, where $\rho_0 = \frac{\rho}{5H}$. In DP-VAPVI, we use $\phi_1, \phi_2, \phi_3, K_1, K_2$[8] to denote the noises we add. For all $\phi_i$, we directly apply Gaussian Mechanism. For $K_i$, in addition to the noise matrix $\frac{1}{\sqrt{2}}(Z + Z^\top)$, we also add $\frac{E}{2}I_d$ to ensure that all $K_i$ are positive definite with high probability (The detailed definition of $E, L$ can be found in Appendix A).

---

**Algorithm 2** Differentially Private Variance-Aware Pessimistic Value Iteration (DP-VAPVI)

---

1: **Input:** Dataset $\mathcal{D} = \{(s_h^\tau, a_h^\tau, r_h^\tau, s_{h+1}^\tau)\}_{\tau,h=1}^{K,H}$ $\mathcal{D}' = \{(\bar{s}_h^\tau, \bar{a}_h^\tau, \bar{r}_h^\tau, \bar{s}_{h+1}^\tau)\}_{\tau,h=1}^{K,H}$. Budget for zCDP $\rho$. Failure probability $\delta$. Universal constant $C$.

2: **Initialization:** Set $\rho_0 \leftarrow \frac{\rho}{5H}$, $\widetilde{V}_{H+1}(\cdot) \leftarrow 0$. Sample $\phi_1 \sim \mathcal{N}\left(0, \frac{2H^4}{\rho_0}I_d\right)$, $\phi_2, \phi_3 \sim \mathcal{N}\left(0, \frac{2H^2}{\rho_0}I_d\right)$,
   $K_1, K_2 \leftarrow \frac{E}{2}I_d + \frac{1}{\sqrt{2}}(Z + Z^\top)$, where $Z_{i,j} \sim \mathcal{N}\left(0, \frac{1}{4\rho_0}\right)$ (i.i.d.), $E = \widetilde{O}\left(\sqrt{\frac{Hd}{\rho}}\right)$. Set $D \leftarrow \widetilde{O}\left(\frac{H^2 L}{\kappa} + \frac{H^4 E \sqrt{d}}{\kappa^{3/2}} + H^3 \sqrt{d}\right)$.

3: **for** $h = H, H-1, \ldots, 1$ **do**

4:      Set $\widetilde{\Sigma}_h \leftarrow \sum_{\tau=1}^K \phi(\bar{s}_h^\tau, \bar{a}_h^\tau)\phi(\bar{s}_h^\tau, \bar{a}_h^\tau)^\top + \lambda I + K_1$

5:      Set $\widetilde{\beta}_h \leftarrow \widetilde{\Sigma}_h^{-1}[\sum_{\tau=1}^K \phi(\bar{s}_h^\tau, \bar{a}_h^\tau) \cdot \widetilde{V}_{h+1}(\bar{s}_{h+1}^\tau)^2 + \phi_1]$

6:      Set $\widetilde{\theta}_h \leftarrow \widetilde{\Sigma}_h^{-1}[\sum_{\tau=1}^K \phi(\bar{s}_h^\tau, \bar{a}_h^\tau) \cdot \widetilde{V}_{h+1}(\bar{s}_{h+1}^\tau) + \phi_2]$

7:      Set $\left[\widetilde{\text{Var}}_h \widetilde{V}_{h+1}\right](\cdot, \cdot) \leftarrow \left\langle \phi(\cdot, \cdot), \widetilde{\beta}_h \right\rangle_{[0,(H-h+1)^2]} - \left[\left\langle \phi(\cdot, \cdot), \widetilde{\theta}_h \right\rangle_{[0,H-h+1]}\right]^2$

8:      Set $\widetilde{\sigma}_h(\cdot, \cdot)^2 \leftarrow \max\{1, \widetilde{\text{Var}}_h \widetilde{V}_{h+1}(\cdot, \cdot)\}$

9:      Set $\widetilde{\Lambda}_h \leftarrow \sum_{\tau=1}^K \phi(s_h^\tau, a_h^\tau)\phi(s_h^\tau, a_h^\tau)^\top / \widetilde{\sigma}_h^2(s_h^\tau, a_h^\tau) + \lambda I + K_2$

10:     Set $\widetilde{w}_h \leftarrow \widetilde{\Lambda}_h^{-1}\left(\sum_{\tau=1}^K \phi(s_h^\tau, a_h^\tau) \cdot \left(r_h^\tau + \widetilde{V}_{h+1}(s_{h+1}^\tau)\right) / \widetilde{\sigma}_h^2(s_h^\tau, a_h^\tau) + \phi_3\right)$

11:     Set $\Gamma_h(\cdot, \cdot) \leftarrow C\sqrt{d} \cdot \left(\phi(\cdot, \cdot)^\top \widetilde{\Lambda}_h^{-1}\phi(\cdot, \cdot)\right)^{1/2} + \frac{D}{K}$

12:     Set $\bar{Q}_h(\cdot, \cdot) \leftarrow \phi(\cdot, \cdot)^\top \widetilde{w}_h - \Gamma_h(\cdot, \cdot)$

13:     Set $\widehat{Q}_h(\cdot, \cdot) \leftarrow \min\{\bar{Q}_h(\cdot, \cdot), H - h + 1\}^+$

14:     Set $\widehat{\pi}_h(\cdot \mid \cdot) \leftarrow \text{argmax}_{\pi_h} \langle \widehat{Q}_h(\cdot, \cdot), \pi_h(\cdot \mid \cdot)\rangle_{\mathcal{A}}$, $\widetilde{V}_h(\cdot) \leftarrow \max_{\pi_h} \langle \widehat{Q}_h(\cdot, \cdot), \pi_h(\cdot \mid \cdot)\rangle_{\mathcal{A}}$

15: **end for**

16: **Output:** $\{\widehat{\pi}_h\}_{h=1}^H$.

---

Below we will show the algorithmic design of DP-VAPVI (Algorithm 2). For the offline dataset, we divide it into two independent parts with equal length: $\mathcal{D} = \{(s_h^\tau, a_h^\tau, r_h^\tau, s_{h+1}^\tau)\}_{\tau \in [K]}^{h \in [H]}$ and $\mathcal{D}' = \{(\bar{s}_h^\tau, \bar{a}_h^\tau, \bar{r}_h^\tau, \bar{s}_{h+1}^\tau)\}_{\tau \in [K]}^{h \in [H]}$. One for estimating variance and the other for calculating $Q$-values.

**Estimating conditional variance.** The first part (line 4 to line 8) aims to estimate the conditional variance of $\widetilde{V}_{h+1}$ via the definition of variance: $[\text{Var}_h \widetilde{V}_{h+1}](s, a) = [P_h(\widetilde{V}_{h+1})^2](s, a) - ([P_h \widetilde{V}_{h+1}](s, a))^2$. For the first term, by the definition of linear MDP, it holds that $\left[P_h \widetilde{V}_{h+1}^2\right](s, a) = \phi(s, a)^\top \int_{\mathcal{S}} \widetilde{V}_{h+1}^2(s') \, d\nu_h(s') = \langle \phi, \int_{\mathcal{S}} \widetilde{V}_{h+1}^2(s') \, d\nu_h(s')\rangle$. We can estimate $\beta_h = \int_{\mathcal{S}} \widetilde{V}_{h+1}^2(s') \, d\nu_h(s')$ by applying ridge regression. Below is the output of ridge regression with raw statistics without noise:

$$\underset{\beta \in \mathbb{R}^d}{\text{argmin}} \sum_{k=1}^K \left[\left\langle \phi(\bar{s}_h^k, \bar{a}_h^k), \beta \right\rangle - \widetilde{V}_{h+1}^2\left(\bar{s}_{h+1}^k\right)\right]^2 + \lambda\|\beta\|_2^2 = \bar{\Sigma}_h^{-1} \sum_{k=1}^K \phi(\bar{s}_h^k, \bar{a}_h^k)\widetilde{V}_{h+1}^2\left(\bar{s}_{h+1}^k\right),$$

where definition of $\bar{\Sigma}_h$ can be found in Appendix A. Instead of using the raw statistics, we replace them with private ones with Gaussian noises as in line 5. The second term is estimated similarly in line 6. The final estimator is defined as in line 8: $\widetilde{\sigma}_h(\cdot, \cdot)^2 = \max\{1, \widetilde{\text{Var}}_h \widetilde{V}_{h+1}(\cdot, \cdot)\}$.[9]

---

[8]We need to add noise to each of the $5H$ counts, therefore for $\phi_1$, we actually sample $H$ i.i.d samples $\phi_{1,h}$, $h = 1, \cdots, H$ from the distribution of $\phi_1$. Then we add $\phi_{1,h}$ to $\sum_{\tau=1}^K \phi(\bar{s}_h^\tau, \bar{a}_h^\tau) \cdot \widetilde{V}_{h+1}(\bar{s}_{h+1}^\tau)^2$, $\forall h \in [H]$. For simplicity, we use $\phi_1$ to represent all the $\phi_{1,h}$. The procedure applied to the other $4H$ statistics are similar.

[9]The $\max\{1, \cdot\}$ operator here is for technical reason only: we want a lower bound for each variance estimate.

**Variance-weighted LSVI.** Instead of directly applying LSVI (Jin et al., 2021), we can solve the variance-weighted LSVI (line 10). The result of variance-weighted LSVI with non-private statistics is shown below:

$$\operatorname*{argmin}_{w\in\mathbb{R}^d} \lambda\|w\|_2^2 + \sum_{k=1}^{K} \frac{\left[\langle\phi(s_h^k,a_h^k),w\rangle - r_h^k - \widetilde{V}_{h+1}(s_{h+1}^k)\right]^2}{\widetilde{\sigma}_h^2(s_h^k,a_h^k)} = \widehat{\Lambda}_h^{-1}\sum_{k=1}^{K}\frac{\phi\left(s_h^k,a_h^k\right)\cdot\left[r_h^k+\widetilde{V}_{h+1}\left(s_{h+1}^k\right)\right]}{\widetilde{\sigma}_h^2(s_h^k,a_h^k)},$$

where definition of $\widehat{\Lambda}_h$ can be found in Appendix A. For the sake of differential privacy, we use private statistics instead and derive the $\widetilde{w}_h$ as in line 10.

**Our private pessimism.** Notice that if we remove all the Gaussian noises we add, our DP-VAPVI (Algorithm 2) will degenerate to VAPVI (Yin et al., 2022). We design a similar pessimistic penalty using private statistics (line 11), with additional $\frac{D}{K}$ accounting for the extra pessimism due to DP.

**Main theorem.** We state our main theorem about DP-VAPVI below, the proof sketch is deferred to Appendix D.1 and detailed proof is deferred to Appendix D due to space limit. Note that quantities $\mathcal{M}_i, L, E$ can be found in Appendix A and briefly, $L = \widetilde{O}(\sqrt{H^3d/\rho})$, $E = \widetilde{O}(\sqrt{Hd/\rho})$. For the sample complexity lower bound, within the practical regime where the privacy budget is not very small, $\max\{\mathcal{M}_i\}$ is dominated by $\max\{\widetilde{O}(H^{12}d^3/\kappa^5),\widetilde{O}(H^{14}d/\kappa^5)\}$, which also appears in the sample complexity lower bound of VAPVI (Yin et al., 2022). The $\sigma_V^2(s,a)$ in Theorem 4.1 is defined as $\max\{1,\operatorname{Var}_{P_h}(V)(s,a)\}$ for any $V$.

**Theorem 4.1.** *DP-VAPVI (Algorithm 2) satisfies $\rho$-zCDP. Furthermore, let $K$ be the number of episodes. Under the condition that $K > \max\{\mathcal{M}_1,\mathcal{M}_2,\mathcal{M}_3,\mathcal{M}_4\}$ and $\sqrt{d} > \xi$, where $\xi :=$* $\sup_{V\in[0,H],\, s'\sim P_h(s,a),\, h\in[H]}\left|\frac{r_h + V(s') - (\mathcal{T}_h V)(s,a)}{\sigma_V(s,a)}\right|$, *for any $0 < \lambda < \kappa$, with probability $1-\delta$, for all policy $\pi$ simultaneously, the output $\widehat{\pi}$ of DP-VAPVI satisfies ($\widetilde{O}$ hides constants and Polylog terms)*

$$v^\pi - v^{\widehat{\pi}} \leq \widetilde{O}\left(\sqrt{d}\cdot\sum_{h=1}^{H}\mathbb{E}_\pi\left[\sqrt{\phi(\cdot,\cdot)^\top\Lambda_h^{-1}\phi(\cdot,\cdot)}\right]\right) + \frac{DH}{K}, \tag{9}$$

*where $\Lambda_h = \sum_{k=1}^{K}\frac{\phi(s_h^k,a_h^k)\cdot\phi(s_h^k,a_h^k)^\top}{\sigma_{\widetilde{V}_{h+1}}^2(s_h^k,a_h^k)} + \lambda I_d$ and $D = \widetilde{O}\left(\frac{H^2L}{\kappa} + \frac{H^4E\sqrt{d}}{\kappa^{3/2}} + H^3\sqrt{d}\right)$.*

*In particular, define $\Lambda_h^\star = \sum_{k=1}^{K}\frac{\phi(s_h^k,a_h^k)\cdot\phi(s_h^k,a_h^k)^\top}{\sigma_{V_{h+1}^\star}^2(s_h^k,a_h^k)} + \lambda I_d$, we have with probability $1-\delta$,*

$$v^\star - v^{\widehat{\pi}} \leq \widetilde{O}\left(\sqrt{d}\cdot\sum_{h=1}^{H}\mathbb{E}_{\pi^\star}\left[\sqrt{\phi(\cdot,\cdot)^\top\Lambda_h^{\star-1}\phi(\cdot,\cdot)}\right]\right) + \frac{DH}{K}. \tag{10}$$

**Comparison to non-private counterpart VAPVI (Yin et al., 2022).** Plugging in the definition of $L, E$ (Appendix A), under the meaningful case that the privacy budget is not very large, $DH$ is dominated by $\widetilde{O}\left(\frac{H^{\frac{11}{2}}d/\kappa^{\frac{3}{2}}}{\sqrt{\rho}}\right)$. According to Theorem 3.2 in (Yin et al., 2022), the sub-optimality bound of VAPVI is for sufficiently large $K$, with high probability, the output $\widehat{\pi}$ satisfies:

$$v^\star - v^{\widehat{\pi}} \leq \widetilde{O}\left(\sqrt{d}\cdot\sum_{h=1}^{H}\mathbb{E}_{\pi^\star}\left[\sqrt{\phi(\cdot,\cdot)^\top\Lambda_h^{\star-1}\phi(\cdot,\cdot)}\right]\right) + \frac{2H^4\sqrt{d}}{K}. \tag{11}$$

Compared to our Theorem 4.1, the additional sub-optimality bound due to differential privacy is $\widetilde{O}\left(\frac{H^{\frac{11}{2}}d/\kappa^{\frac{3}{2}}}{\sqrt{\rho}\cdot K}\right) = \widetilde{O}\left(\frac{H^{\frac{11}{2}}d/\kappa^{\frac{3}{2}}}{\epsilon\cdot K}\right)$[10] In the most popular regime where the privacy budget $\rho$ or $\epsilon$ is a constant, the additional term due to differential privacy also appears as a lower order term.

**Instance-dependent sub-optimality bound.** Similar to DP-APVI (Algorithm 1), our DP-VAPVI (Algorithm 2) also enjoys instance-dependent sub-optimality bound. First, the main term in (10) improves PEVI (Jin et al., 2021) over $O(\sqrt{d})$ on feature dependence. Also, our main term admits no explicit dependence on $H$, thus improves the sub-optimality bound of PEVI on horizon dependence. For more detailed discussions about our main term, we refer readers to (Yin et al., 2022).

---

[10]Here we apply the second part of Lemma 2.6 to achieve $(\epsilon,\delta)$-DP, the notation $\widetilde{O}$ also absorbs $\log\frac{1}{\delta}$ (only here $\delta$ denotes the privacy budget instead of failure probability).

## 5 SIMULATIONS

In this section, we carry out simulations to evaluate the performance of our DP-VAPVI (Algorithm 2), and compare it with its non-private counterpart VAPVI (Yin et al., 2022) and another pessimism-based algorithm PEVI (Jin et al., 2021) which does not have privacy guarantee.

**Experimental setting.** We evaluate DP-VAPVI (Algorithm 2) on a synthetic linear MDP example that originates from the linear MDP in (Min et al., 2021; Yin et al., 2022) but with some modifications.[11] For details of the linear MDP setting, please refer to Appendix F. The two MDP instances we use both have horizon $H = 20$. We compare different algorithms in figure 1(a), while in figure 1(b), we compare our DP-VAPVI with different privacy budgets. When doing empirical evaluation, we do not split the data for DP-VAPVI or VAPVI and for DP-VAPVI, we run the simulation for 5 times and take the average performance.

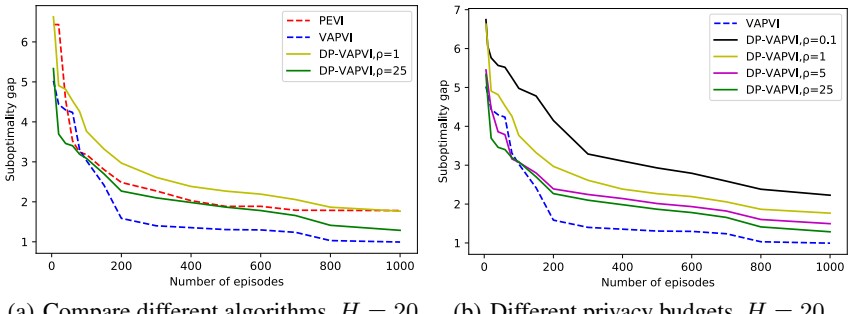

(a) Compare different algorithms, $H = 20$    (b) Different privacy budgets, $H = 20$

Figure 1: Comparison between performance of PEVI, VAPVI and DP-VAPVI (with different privacy budgets) under the linear MDP example described above. In each figure, y-axis represents sub-optimality gap $v^\star - v^{\widehat{\pi}}$ while x-axis denotes the number of episodes $K$. The horizons are fixed to be $H = 20$. The number of episodes takes value from 5 to 1000.

**Results and discussions.** From Figure 1, we can observe that DP-VAPVI (Algorithm 2) performs slightly worse than its non-private version VAPVI (Yin et al., 2022). This is due to the fact that we add Gaussian noise to each count. However, as the size of dataset goes larger, the performance of DP-VAPVI will converge to that of VAPVI, which supports our theoretical conclusion that the cost of privacy only appears as lower order terms. For DP-VAPVI with larger privacy budget, the scale of noise will be smaller, thus the performance will be closer to VAPVI, as shown in figure 1(b). Furthermore, in most cases, DP-VAPVI still outperforms PEVI, which does not have privacy guarantee. This arises from our privitization of variance-aware LSVI instead of LSVI.

## 6 CONCLUSION AND FUTURE WORKS

In this work, we take the first steps towards the well-motivated task of designing private offline RL algorithms. We propose algorithms for both tabular MDPs and linear MDPs, and show that they enjoy instance-dependent sub-optimality bounds while guaranteeing differential privacy (either zCDP or pure DP). Our results highlight that the cost of privacy only appears as lower order terms, thus become negligible as the number of samples goes large.

Future extensions are numerous. We believe the technique in our algorithms (privitization of Bernstein-type pessimism and variance-aware LSVI) and the corresponding analysis can be used in online settings too to obtain tighter regret bounds for private algorithms. For the offline RL problems, we plan to consider more general function approximations and differentially private (deep) offline RL which will bridge the gap between theory and practice in offline RL applications. Many techniques we developed could be adapted to these more general settings.

---

[11]We keep the state space $\mathcal{S} = \{1, 2\}$, action space $\mathcal{A} = \{1, \cdots, 100\}$ and feature map of state-action pairs while we choose stochastic transition (instead of the original deterministic transition) and more complex reward.

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
