# OpenReview forum: "Offline Reinforcement Learning with Differential Privacy"
_ICLR.cc/2023/Conference — Submitted to ICLR 2023_

### Official Review · Reviewer_MbLZ · 2022-10-19

**Confidence:** 3
**Correctness:** 3
**Technical Novelty And Significance:** 4
**Empirical Novelty And Significance:** 1
**Recommendation:** 5

**Clarity, Quality, Novelty And Reproducibility:**

Overall, the paper is clearly-written, but some improvements are needed.

The presentation of the key components can be improved. For example, it takes me some time to understand the optimization problem in (3), so I think the paper would be better if some descriptions are provided to explain what is happening in (3) and how is $E_P$ chosen (a short sentence suffices). I think this is the non-trivial part to combine common DP mechanisms in pessimism-based VI algorithm, so it is definitely worth spending some space to explain it more clearly.

I have some other questions:
> Our algorithms involve substantial technical innovation over previous works on online DP-RL with joint DP guarantee.

For the tabular case, is the main novelty how you obtain private visitation counts that satisfies $n’_{s,a} = \sum_{s’} n’_{s,a,s’}$? For the linear MDP case, the proposed algorithm seems like a straightforward application of Gaussian mechanism in the plug-in method. Am I missing something here?

> Assumption 2.1 is the weakest assumption needed for accurately learning the optimal value.

"weakest" in what sense? Do you have any citation for that?

For Algorithm 2, you add a noise with variance of the order of $H^4$. I am wondering if this can be an issue in environments with a higher horizon (e.g., 100 or 1000). Also where does the quadruple dependence come from? Is it improvable?


**Strength And Weaknesses:**

**Strength**
- The paper proposes a new offline RL algorithm with DP, which is claimed to be the first DP RL algorithm in the offline setting. I think the paper can have significant contribution to the offline RL community.
- The paper in general is clearly-written and easy to follow.
- The theoretical analysis is sound (although I acknowledge that I only check the proof for tabular case, and didn’t check the proof for linear MDP carefully).

**Weaknesses**
- The definition of DP for the RL setting used in the paper is not well-motivated (see comments below).
- The presentation of the key components in Algorithm 1 and Algorithm 2 is not very clear to me (see the next part on quality and clarity).
- Major improvement is required for the empirical study.

**DP Definition**
The paper defines neighboring datasets as two datasets that differ in one trajectory. However, I can think of other definitions in the RL setting. For example, two datasets that differ in one transition, or differ in one observed reward (if we want to preserve the privacy for reward only). The paper should provide more motivation for this particular definition used here, and discuss how existing work define DP in the RL setting.

**Empirical study**
I think the empirical study requires some improvements. The empirical study is supposed to support the theocratical findings, but I think the current results raise more concerns.

First of all, the results do not have error bars, so I don’t know whether the difference are statistically significant. Please include the error bar in the results. Moreover, the paper mentions
> as the size of dataset goes larger, the performance of DP-VAPVI will converge to that of VAPVI

I don’t see the performance of DP-VAPVI converges to the non-private one in Figure 1. Maybe try to run the experiments with more episodes. Otherwise, I think this observation is not true.

Finally, please explain how significant the difference in performance between the private and non-private ones are. For example, non-private gets a suboptimality gap ~= 1, while the private one with $\rho = 5$ gets a gap ~= 2. Is the difference (i.e., 1) significant? What is the range here? What value of $\rho$ is commonly used in practice?


**Summary Of The Paper:**

The paper proposes offline RL algorithms with DP guarantees for tabular case and linear MDP. The key components are the private estimates of the visitation counts/conditional variance based on the Gaussian Mechanism, with some modifications to allow the use of existing pessimism-based value iteration algorithms. The paper provides privacy guarantees and utility guarantees for the proposed algorithms, and an empirical study to support the theoretical findings.

**Summary Of The Review:**

In summary, I am giving a weak accept recommendation, conditional on the empirical section can be improved as suggested. I think the paper can have a significant contribution to the offline RL community as a first step towards more practical DP offline RL algorithms, however, some improvements are required as mentioned in my comments on weakness and clarity. I would be happy to raise my score if the concerns are addressed.

------Post-rebuttal update------
After the author response and internal discussions with other reviewers and AC, I think my concerns about the presentation clarity and the empirical study have not been fully addressed. I think these issues are not difficult to address, however, the authors did not upload a revision. Therefore, I think the paper requires some improvements before being published, and I slightly lowered my score to 5. However, I encourage the authors to keep improving the paper and re-submit to a future conference.

---

> ### Author Response · Authors · 2022-11-13
> **Response to Reviewer MbLZ (Part 2)**
>
> >For the tabular case, is the main novelty how you obtain private visitation counts that satisfies $n^\prime_{s,a}=\sum_{s^\prime}n^\prime_{s,a,s^\prime}$? For the linear MDP case, the proposed algorithm seems like a straightforward application of Gaussian mechanism in the plug-in method. Am I missing something here?
>
> For the tabular MDP case, the condition is only a middle step towards privatization of Bernstein-type pessimism. Under tabular MDP, the best known work on RL with joint DP ([1]) only privatizes Hoeffding-type bonus, where noises are added to visitation numbers and they only need to deal with bonuses like $b^k_h(s,a)=\frac{H}{\sqrt{\widetilde{N}^k_h(s,a)}}$ where $\widetilde{N}$ is the noisy version of visitation number. In comparison, we privatize Bernstein-type pessimism which could provide tighter results while it also raises technical challenges. We managed to provide a confidence bound using private pessimism and lower order additional terms, prove its validity and bound this private Bernstein pessimism by its non-private counterpart in our final result. All such techniques are novel to our knowledge. Similarly, under linear MDP, although we only apply Gaussian mechanism, it is non-trivial to provide a bound for suboptimality since we are the first to privatize weighted LSVI. In comparison, the best known algorithm ([2]) under this setting only privatize LSVI, which is easier to handle. Furthermore, we believe our techniques can be further applied under online settings for tighter regret bounds.
>
> >"weakest" in what sense? Do you have any citation for that?
>
> We are sorry for not clarifying this. Weakest here means that to accurately learn the optimal policy, the plug in policy should at least trace the (state,action) space of one optimal policy. This statement is originally stated in [3], where the authors showed that when this assumption is not met, a constant gap is suffered.
>
> >For Algorithm 2, you add a noise with variance of the order of $H^4$. I am wondering if this can be an issue in environments with a higher horizon (e.g., 100 or 1000). Also where does the quadruple dependence come from? Is it improvable?
>
> We agree that this could be an issue when horizon is huge, while the large variance will not change the fact that the additional cost is a lower order term. Currently, we do not think it is improvable, since the calculation for $\widetilde{\beta}$ includes summation of $\widetilde{V}^2$ and the l2 sensitivity is of order $H^2$.
>
> Thanks again for the helpful review! We will improve the current version based on your suggestions. Please let us known whether we have answered all your questions clearly and we are happy to further discuss about this paper.
>
> [1] Sayak Ray Chowdhury and Xingyu Zhou. Differentially private regret minimization in episodic markov decision processes. arXiv preprint arXiv:2112.10599, 2021.
>
> [2] Dung Daniel Ngo, Giuseppe Vietri, and Zhiwei Steven Wu. Improved regret for differentially private
> exploration in linear mdp. arXiv preprint arXiv:2202.01292, 2022.
>
> [3] Ming Yin and Yu-Xiang Wang. Towards instance-optimal offline reinforcement learning with pessimism. Advances in neural information processing systems, 34, 2021.

---

> > ### Comment · Reviewer_MbLZ · 2022-12-06
> > **Response to Authors**
> >
> > Thank you for your detailed response. I updated my review based on internal discussions with other reviewers. Please see the updated review for my final assessment. Thank you.

---

> ### Author Response · Authors · 2022-11-13
> **Response to Reviewer MbLZ (Part 1)**
>
> Thanks for the high-quality and detailed review. This is the most helpful review we have ever received. We will try to address your concerns below.
>
> >DP Definition. The paper defines neighboring datasets as two datasets that differ in one trajectory. However, I can think of other definitions in the RL setting. For example, two datasets that differ in one transition, or differ in one observed reward (if we want to preserve the privacy for reward only). The paper should provide more motivation for this particular definition used here, and discuss how existing work define DP in the RL setting.
>
> We provide one example in the first paragraph of introduction and we restate it here. Consider a dataset of medical treatment, the dataset consists of $n$ patients and the data for each patient includes the (health condition, treatment, treatment outcome) for $H$ times. To be more specific, for each patient, at each time step, the patient reports her health condition (age, disease, etc.), then the doctor decides the treatment (which medicine to use, the dosage of medicine, etc.), finally there is treatment outcome (whether the patient feels good, etc.) and the patient transitions to another health condition. Here, (health condition, treatment, treatment outcome) corresponds to (state, action, reward) and the dataset can be considered as $n$ trajectories sampled from a MDP with horizon $H$. Offline RL aims to learn a good policy to decide treatment given health condition. For this specific offline RL task, Differential Privacy can protect the data of any single patient from being identified by the output policy. Current DP-RL works mainly focus on online RL where there are definitions like Joint DP and Local DP (as discussed in Appendix B). Both definitions characterize each user as one trajectory and aim to protect the information of each trajectory, which is consistent with our definition of DP.
>
> >About empirical study.
>
> Since we are taking the first step towards differential privacy under offline RL, we mainly analyze our algorithms through theory while only running simulations on toy examples. We will include more complex experiments and improve the current ones in the next version.
>
> >About clarity.
>
> Thanks for your suggestions. We totally agree and will improve our writing in the next version.

---

### Official Review · Reviewer_txBL · 2022-10-22

**Confidence:** 3
**Correctness:** 4
**Technical Novelty And Significance:** 3
**Empirical Novelty And Significance:** 2
**Recommendation:** 5

**Clarity, Quality, Novelty And Reproducibility:**

* Quality: In the interest of time, I did not check the proof. But, according to a sketch of the analysis, it sounds correct.
* Clarity: Yes, it is clear
* Novelty: Incremental novelty over previous offline works by adding DP. It adds  to some extent  to the knowledge of the community.
* Reproducibility: Not clear to me, since source code and dataset were not available in supplementary materials.

**Strength And Weaknesses:**

Strength:
* Addresses an important problem in offline RL
* The paper is well organized, theoretically grounded

Weakness:
* Lack of empirical evaluation such as experimentations on benchmark tasks such as D4RL. Additionally is there any reason that there is no simulation results for DP-APVI algorithm ?
* Incremental Novelty. Would be nice to distinguish more clearly the novelty of this paper from previous offline works (i.e., VAPVI).


**Summary Of The Paper:**

This paper proposes two privacy-preserved (i.e., DP Guaranteed) versions of existing offline RL algorithms (i.e., APVI and VAPVI) proposed in previous works . DP-APVI  is for tabular settings and  DP-VAPVI  for the case with linear function approximation (under linear MDP assumption). The authors provide theoretical analysis and also simulation results to confirm their theoretical analysis and show that their DP Guaranteed algorithms come with little drop in utility.

**Summary Of The Review:**

Overall, the paper is well motivated, aiming to solve an important problem in the area of offline reinforcement learning. The work is theoretically well grounded, and well organized. As said the novelty is low, and I think it lacks enough experimental evaluation.

---

> ### Author Response · Authors · 2022-11-13
> **Response to Reviewer txBL**
>
> Thanks for your high-quality review and the positive score. We will reply to the weaknesses you stated.
>
> >Lack of empirical evaluation such as experimentations on benchmark tasks such as D4RL. Additionally is there any reason that there is no simulation results for DP-APVI algorithm ?
>
> Since we are taking the first step towards differential privacy under offline RL, we mainly analyze our algorithms through theory while only running simulations on toy examples. We will include more complex experiments in the next version.
>
> >Incremental Novelty. Would be nice to distinguish more clearly the novelty of this paper from previous offline works (i.e., VAPVI).
>
> To the best of our knowledge, most works of DP-RL built upon existing algorithms / analysis in (non-private) RL and the modifications are about appropriately adding noises to guarantee DP (please refer to our Appendix B). However, even given a base algorithm, making it differentially private is often not trivial. Take our DP-APVI as an instance, although the proof schedule follows that of APVI, it requires nontrivial techniques to handle Bernstein type pessimism. We managed to provide a confidence bound using private pessimism and lower order additional terms, prove its validity and bound this private Bernstein pessimism by its non-private counterpart in our final result. All such techniques are novel to our knowledge. We also include tricks such as solving the optimization problem (Eq (3)) to further improve the parameters in the lower order term. I hope you will agree that the technical part in this paper is not trivial.
>
> Thanks again for the helpful review! We hope that our response could address your main concerns about lack of novelty. Please let us known whether we have answered all your questions clearly and we are happy to further discuss about this paper.

---

### Official Review · Reviewer_g4wC · 2022-10-31

**Confidence:** 4
**Correctness:** 3
**Technical Novelty And Significance:** 2
**Empirical Novelty And Significance:** Not applicable
**Recommendation:** 3

**Clarity, Quality, Novelty And Reproducibility:**

Originality:  the work is original in that this particular setting for differential privacy has not been considered before.  The algorithms themselves are heavily based on previous work of Yin and Want 2021b.
Clarity:  As mentioned above the paper lacks in clarity.

**Strength And Weaknesses:**

Strengths
- the problem is interesting and has good motivation:  protection of sensitive information in the trajectories saved in offline reinforcement learning, and this is especially important in healthcare and financial contexts.

Weaknesses
1.  The paper is not clearly written.  Objects and variables are used without proper notation or introduction.  For instance I'm assuming $n$ is the number of trajectories in the dataset, but this doesn't seem to be stated.  What is $\phi(s,a)$, used in algorithm 2?
Also, the theorems can be stated more clearly.  For instance Theorem 3.1 has parentheses with phrases that distract from the reading of the theorem - this should be re-written.
2.  It's not obvious how the noise value $\sigma^2$ is derived for Algorithm 1.  Shouldn't the l2 sensitivity of the counts be $Hn\sqrt{n}$, and so $\sigma^2= H^2n^3/(2\rho)$ ?
3.  Too much information is relegated to the appendix, to the extent that it isn't easy to the read the paper without reading the appendix.

**Summary Of The Paper:**

The paper provides an algorithm for Differentially-Private offline reinforcement learning, for the tabular and linear MDP settings.

**Summary Of The Review:**

While the topic of the paper is important, has good motivation, and is novel, the paper is very difficult to read and may contain errors.

---

> ### Author Response · Authors · 2022-11-13
> **Response to Reviewer g4wC**
>
> Thanks for your review. we believe there is some misunderstanding and we will try to address your concerns below.
>
> >Objects and variables are used without proper notation or introduction.
>
> We politely disagree and the two examples you mentioned both have been defined. For the sample complexity $n$, we assume the dataset is $\mathcal{D}=\{(s_{h}^{\tau}, a_{h}^{\tau}, r_{h}^{\tau}, s_{h+1}^{\tau})\}_{\tau\in[n]}^{h \in[H]}$ and the subscript means there are $n$ trajectories in total. The same notation is also used in the introduction of the offline RL setting in Section 2. For $\phi(s,a)$ in Algorithm 2, it is the feature map of linear MDP, which has been clearly defined in Section 2. For linear MDP setting, we recommend [Jin et. al.], which could serve as a good introduction to the setting.
>
> >It's not obvious how the noise value $\sigma^2$ is derived for Algorithm 1. Shouldn't the l2 sensitivity of the counts be $Hn\sqrt{n}$, and so $\sigma^2=H^2n^3/(2\rho)$?
>
> For the visitation numbers of all (state,action,time step) pairs (i.e. {$n_{s_h,a_h}$}), since two neighboring datasets $\mathcal{D}$ and $\mathcal{D}^\prime$ differ by just one trajectory, {$n_{s_h,a_h}$} with respect to $\mathcal{D}$ and $\mathcal{D}^\prime$ only differ in at most $2H$ elements and the difference is at most 1. Therefore the $\ell_2$ sensitivity of {$n_{s_h,a_h}$} is $\sqrt{2H}$ and the choice of $\sigma^2$ follows directly. Similar analysis apply to {$n_{s_h,a_h,s_{h+1}}$}. Since all visitation numbers are at most $n$, we do not understand where does the $n\sqrt{n}$ come from. If you have further concerns about correctness, please provide your reason.
>
> >Too much information is relegated to the appendix, to the extent that it isn't easy to the read the paper without reading the appendix.
>
> We leave the proof and proof sketches to Appendix due to space limit. In the next version, we will improve the writing for readability.
>
> [Jin et. al.] Chi Jin, Zhuoran Yang, Zhaoran Wang, and Michael I Jordan. Provably efficient reinforcement learning with linear function approximation. In Conference on Learning Theory, pp. 2137–2143. PMLR, 2020.

---

### Decision · Program_Chairs · 2023-01-20

**Decision:**

Reject

**Justification For Why Not Higher Score:**

The paper has several shortcomings summarized in the meta review. There was a general consensus among reviewers that the paper is below bar for acceptance.

**Justification For Why Not Lower Score:**

N/A

**Metareview: Summary, Strengths And Weaknesses:**

The paper studies differentially private (DP) reinforcement learning (RL) in the offline setting. The authors provide DP algorithms for this problem and formally prove their privacy and utility guarantees. The utility guarantees are in terms of learning bounds under the tabular and linear MDP settings.

Despite introducing some novel techniques such as privately computing the conditional variance, the analysis seems to largely follow existing works, e.g., [Chowdhury & Zhou 2021] and [Yin & Wang 2021].

The reviewers have also pointed out that the algorithmic novelty is limited as the main algorithm (Algorithm 1) relies heavily on a prior work [Yin & Wang 2021].

There have been also some concerns regarding the tightness of the upper bounds and the possibility of simplifying the algorithm by using the discrete Gaussian mechanism. Additionally, there is some criticism about the lack of motivation behind the notion of "neighboring datasets" in the proposed definition of DP in offline RL.

Moreover, the presentation and writing could be improved in several parts of the paper, especially in Sec. 2.

Finally, the paper could have benefited from a more convincing, extensive empirical evaluation on benchmark datasets for RL.

All in all, the paper can benefit from a careful revision to address these issues.